# Antioxidant and Anti-Inflammatory Mechanisms of Cardamonin through Nrf2 Activation and NF-kB Suppression in LPS-Activated BV-2 Microglial Cells

**DOI:** 10.3390/ijms241310872

**Published:** 2023-06-29

**Authors:** Kimberly Barber, Patricia Mendonca, Jasmine A. Evans, Karam F. A. Soliman

**Affiliations:** 1Division of Pharmaceutical Sciences, College of Pharmacy and Pharmaceutical Sciences, Institute of Public Health, Florida A&M University, Tallahassee, FL 32307, USA; kimberly1.barber@famu.edu (K.B.); jasmine2.evans@famu.edu (J.A.E.); 2Department of Biology, College of Science and Technology, Florida A&M University, Tallahassee, FL 32307, USA

**Keywords:** cardamonin, microglia, neurodegeneration, neuroinflammation, oxidative stress

## Abstract

Chronic oxidative stress (OS) and inflammation are implicated in developing and progressing neurodegenerative diseases (NDs). The chronic activation of microglia cells leads to the overproduction of several substances, including nitric oxide and reactive oxygen species, which can induce neurodegeneration. Natural compounds have recently been investigated for their potential to protect cells from OS and to improve many disease-related conditions. Cardamonin (CD) is a bioactive compound in many plants, such as *Alpinia katsumadai* and *Alpinia conchigera.* The present study examined the effects of CD on LPS-activated BV-2 microglial cells. The cell viability results showed that the increasing concentrations of CD, ranging from 0.78 to 200 µM, induced BV-2 cell cytotoxicity in a dose–response manner. In the nitric oxide assay, CD concentrations of 6.25 to 25 µM reduced the release of nitric oxide in LPS-activated BV-2 cells by 90% compared to those treated with LPS only (*p* ≤ 0.0001). CD (6.25 µM) significantly decreased the cellular production of SOD (3-fold (*p* ≤ 0.05)) and increased the levels of expression of CAT (2.5-fold (*p* ≤ 0.05)) and GSH (2-fold (*p* ≤ 0.05)) in the LPS-activated BV-2 cells. Furthermore, on RT-PCR arrays, CD (6.25 µM) downregulated mRNA expression of CCL5/RANTES (5-fold), NOS2 (2-fold), SLC38A1 (3-fold), TXNIP (2-fold), SOD1 (2-fold), SOD2 (1.5-fold) and upregulated GSS (1.9-fold), GCLC (1.7-fold) and catalase (2.9-fold) expression, indicating CD efficacy in modulating genes involved in OS and inflammation. Furthermore, CD (6.25 µM) increased the expression of nuclear factor erythroid 2-related factor 2 (Nrf2) and lowered the levels of Kelch-like ECH-associated protein 1 (Keap1), indicating that this may be the signaling responsible for the elevation of antioxidant factors. Lastly, the results showed that CD (6.25 µM) modulated genes and proteins associated with the NF-kB signaling, downregulating genes related to excessive neuroinflammation. These results imply that CD may be a potential compound for developing therapeutic and preventive agents in treating neurodegeneration induced by excessive OS and inflammation.

## 1. Introduction

Alzheimer’s disease (AD) is the most common neurodegenerative disease affecting the elderly. Patients with AD have gradual decreases in memory and other cognitive functions, which eventually lead to neuronal death. Oxidative stress (OS) is a crucial factor contributing to the onset and progression of AD. However, the mechanisms that lead to the disruption of redox balance and the sources of free radicals remain unknown. Microglia-derived excessive OS has been implicated in Amyloid β (Aβ) and neurofibrillary tangles (NFTs)-induced neurotoxicity. Evidence has suggested that OS may increase the production and collection of Aβ and facilitate the phosphorylation and polymerization of NFTs, forming a dangerous cycle that aids the initiation and progression of AD [1]. Alzheimer’s disease incidence increased by 16% in the United States during the COVID-19 pandemic and currently, one in three elderly dies from AD or dementia. It kills more than breast cancer and prostate cancer combined [2]. Therefore, finding a therapy to control and reduce the rate of AD incidence is crucial.

Even though there is no known cure for neurodegenerative diseases (NDs), there are many therapy options, most of which are symptomatic. Over the past few decades, interest in naturally occurring phytochemical compounds with antioxidant, anti-inflammatory and neuroprotective potential has increased. Natural compounds are more affordable and easily accessible in the ingestible form [3] and have been safe in treating diseases for thousands of years [4].

Cardamonin (CD) (2,4-dihydroxy-6-methoxychalcone) is an example of a natural chalcone found in medicinal herbs that have been used extensively to treat digestive-system-related diseases for thousands of years. Chalcones are aromatic enones (1,3,-diphenyl -2-propanone core structure) of the flavonoid family. CD has shown anti-inflammatory, antioxidative, anticancer and vasorelaxant activities (4). It is a chalconoid isolated from several plants, including *Alpinia katsumadai*, *Alpinia conchigera* and *Alpinia* gagnepainii [5]. Chalcones are named so because of their yellow pigmentation and are found in plants’ petals, fruits, leaves, heartwood, bark and roots [6]. CD has been demonstrated to suppress the production of proinflammatory cytokines, such as tumor necrosis factor, in activated macrophages [7]. The anti-inflammatory properties of some chalcones are recognized by their ability to inhibit the expression of the inducible enzymes nitric oxide synthase (iNOS) and cyclooxygenase-2 (COX-2) and thus the generation of nitric oxide (NO) and prostaglandin E2. The anti-inflammatory activity of CD has also been associated with its effect on the modulation of the nuclear factor-κB (NF-κB). This protein complex is critical in regulating immune responses to infection [6]. Recent studies have reported CD’s antioxidant and anti-inflammatory properties, interacting with cellular targets such as Nrf2 [7,8]. CD has also been shown to stop the activation and production of matrix metalloproteinases, suppress the NF-κB signaling pathway and raise the levels of the antioxidant proteins heme oxygenase-1 and NADPH quinone oxidoreductase 1, leading to antioxidant effects [8]. The present study investigated CD’s potential to reduce OS and inflammation in activated BV-2 microglia cells. CD’s molecular mechanism in modulating oxidative-stress-associated genes and proteins was examined, as well as CD’s role in activating the Nrf2/Keap1 and NF-κB signaling pathways, which play a critical role in the development and progression of neurodegeneration.

## 2. Results

### 2.1. The Effect of CD and LPS on BV-2 Cell Viability and Nitric Oxide Production

The viability of BV-2 cells treated with CD and LPS was assessed using Alamar Blue assay after 24 h. CD caused a concentration-dependent decrease in cell viability in concentrations of 6.25 µM and over (Figure 1). Different concentrations of LPS ranged from 0.125 to 1 µg/mL to establish an optimum dose. The concentrations tested did not show any statistically significant difference; all cells were 100% alive (Figure 2). Based on these results, the concentrations of 6.25 µM of CD and 1 µg/mL of LPS were selected for subsequent investigations. 

The investigation of the effects of CD on nitric oxide production in LPS-activated BV-2 microglial cells showed that LPS increased nitrite expression compared to the control. When cells were treated with different concentrations of CD (0.78–25 µM), there was a significant decrease in nitrite expression, almost returning to the control level in the highest concentrations (Figure 3).

### 2.2. Effect of CD on Catalase, Glutathione and Superoxide Dismutase Expression

The effect of CD was investigated on the expression of catalase, glutathione and superoxide dismutase. CD significantly increased catalase activity (Figure 4A) and glutathione expression (Figure 4B) compared to the control group, but LPS did not show any statistically significant upregulation. However, the combination of CD and LPS induced a significant increase in catalase activity and glutathione expression compared to the control levels. The effects of CD on superoxide dismutase expression (Figure 4C) in BV-2 microglial cells showed that CD only had no significant effect on SOD production compared to the control group, but when cells were treated with LPS, there was a significant increase in SOD production. Treating the cells with CD and subsequent activation with LPS after 1 h reduced the expression of SOD almost to control levels.

### 2.3. Effects of CD on Oxidative Stress PCR Arrays

PCR arrays were utilized to evaluate the effects of CD on specific genes. For all the experiments, the cells were treated with CD for 1 h and then stimulated by LPS. The results of these combined treatments showed that CD significantly upregulated catalase (2.9-fold) (Figure 5A), an important antioxidant enzyme that helps to minimize oxidative stress by removing cellular hydrogen peroxide. It also increased the expression of GSS (1.9-fold) (Figure 5B), an essential gene for a variety of biological functions, including cell defense from free radical oxidative damage, xenobiotic detoxification and membrane transport. CD also increased GCLC mRNA expression (1.7-fold) (Figure 5C), a key component of a long-term adaptive system that allows cells to function under severe oxidative stress for lengthy periods. Moreover, results showed that CD downregulated mRNA expression of *CCL5/RANTES* (5-fold) (Figure 6A), promoting inflammatory cell recruitment and activation. We also observed a reduction in the expression of SLC38A1 (3-fold) (Figure 6B), which is increased during oxidative stress, and NOS2 (2-fold) (Figure 6C), a reactive free radical associated with a range of biological processes such as antibacterial activity and anticancer activity. Additionally, CD also reduced the expression of TXNIP (2-fold) (Figure 7A), which regulates thioredoxin, an essential redox protein that controls reactive oxygen species (ROS) levels in cells, SOD1 (2-fold) (Figure 7B), an internal antioxidant enzyme that controls the amount of oxidative stress produced by mitochondrial and cytosolic superoxide and SOD2 (1.5-fold) (Figure 7C), which is engaged in cellular systems to repair oxidative damage. The data showed CD efficacy in modulating genes involved in inflammation and modulation of ROS levels.

### 2.4. The Effect of CD in Nrf2 and Keap1 Genes and Proteins Expression

RT-PCR assays were performed to evaluate the effect of CD on the expression of Nrf2 and Keap1. The increase in the expression of Nrf2 was statistically significant after CD treatment compared to the control cells. However, the highest expression was observed when cells were treated with CD and then stimulated with LPS after 1 h, resulting in a 10-fold increase in Nrf2 expression (Figure 8). The opposite effect was observed on Keap1 expression, where CD reduced the expression of Keap1 compared to the control. When CD and LPS were combined, there was also a reduction in Keap1 expression, showing that even in the presence of LPS, CD was able to decrease the expression of Keap1 (Figure 9).

Specific antibodies for Nrf2 and Keap1 were used to validate the PCR results at the protein level. Results showed that CD treatment increased the levels of Nrf2 protein expression compared to the control. When the CD treatment was followed by LPS stimulation after 1 h, there was an even higher increase in Nrf2 protein expression levels (Figure 10). On the other hand, CD had the opposite effect on the expression of the Keap1 protein, reducing the levels observed in the control. The same was observed when CD and LPS were combined (Figure 10). These data validate the findings of the CD effect on the transcription level.

### 2.5. The Effect of NFκB Signaling on Associated Genes and Proteins

Real-time PCR using individual primers was performed to investigate the effect of CD on the mRNA expression of several genes associated with NF-κB signaling. The results showed that when cells were treated with CD and then stimulated by LPS after 1 h, CD downregulated mRNA expression of NF-κB1 (6-fold) (Figure 11A), NF-κB2 (2-fold) (Figure 11B) and IKBκB (10-fold) (Figure 11C). NF-κB1 and NF-κB2 are crucial in regulating the immune system’s reaction to infections and play an essential role in how cells respond to oxidative stress and inflammation. IKBκB is an enzyme complex component of the NF-κB signaling pathway. Furthermore, CD upregulated NFκBIA mRNA expression (2.6-fold) (Figure 11D), which blocks the nuclear localization signals of REL dimers, trapping them in the cytoplasm and inhibiting the activity of dimeric NF-κB/REL complexes. The combined treatment did not cause any statistically significant effect on the mRNA expression of NIK (Figure 11E) or RELA (Figure 11F).

Western analysis was used to confirm the RT-PCR results at the protein level using specific antibodies against NF-κB1, NF-κB2, IKBκB and RELA. The findings demonstrated that LPS increased the expression of these proteins in BV-2 microglial cells but that this induction was significantly decreased when CD was added 1 h before LPS treatment (Figure 12A–E). The results were consistent with the data obtained from the RT-PCR assays.

## 3. Discussion

Alzheimer’s disease is a progressive neurodegenerative disorder characterized by cognitive and memory deterioration caused by neuronal death influenced by extracellular Aβ deposits and the formation of intracellular neurofibrillary tangles. AD brain studies have shown that activation of glial cells also contributes to the production of excessive quantities of free radicals, cytokines and NO, which could damage neuronal cells [9]. Oxidative stress is an important neurodegeneration characteristic contributing to microglial activation [10]. An imbalance between pro-oxidants and antioxidants results when the endogenous antioxidant system cannot precisely remove the free radical produced. ROS and free radicals promote OS in healthy cells by damaging DNA, RNA, lipids and proteins. Therefore, antioxidants play a crucial role in cellular defense against ROS and free radicals and in inhibiting neurodegenerative diseases [10].

Considering the role of activated microglia in neurodegeneration, the current study investigated CD antioxidant and anti-inflammatory properties on BV-2 microglial cells. The obtained results show that CD treatment induced a 90% inhibition of NO production in the LPS-activated cells, indicating that CD may have the potential to decrease the levels of inflammation and oxidative stress caused by high production of NO. The exposure of the BV-2 microglial cells to LPS for 24 h increased proinflammatory cytokines and ROS production. Increasing the production of ROS from infiltrating immune cells is very harmful and destructive to cells because ROS causes the production of peroxynitrite from the combination of NO and superoxide, which are potent mediators of lipid peroxidation in the brain, inflammation, neuronal death and neurodegeneration [11].

Further evaluation of the CD effects on the LPS-activated BV-2 cells showed that CD increased CAT and GSH levels. GSH is the most abundant nonprotein thiol tripeptide in the CNS that acts as a key cellular antioxidant protecting the brain neurons [12]. GSH deficiency has been detected in aging and various pathologies, including NDs. On the other hand, the expression of SOD was downregulated by pretreatment with CD. SOD catalyzes the dismutation of the superoxide anion radical into molecular oxygen or hydrogen peroxide. Then, CAT catalyzes the decomposition of hydrogen peroxide into water and oxygen. These enzymes work together to help reduce the levels of ROS and keep the cell at healthy levels of OS. The data showed that CD treatment increased GSH and CAT levels in the activated microglia, which could help lower levels of OS in the cells [12], even though CD did not increase the expression of SOD in our model.

RT-PCR profiling arrays for genes associated with OS confirmed CD’s antioxidant protection on BV-2 microglial cells, where specific genes were modulated. Results revealed that CD reduced the mRNA expression of CCL5/RANTES, which stimulates the recruitment and activation of inflammatory cells [13], and NOS2, which is responsible for the production of NO, a reactive free radical connected to various biological processes [13]. The inhibition of NOS2 may explain the reduction in NO production observed in this study. There was also a decrease in the expression of TXNIP (2-fold), which controls thioredoxin, a crucial redox protein that regulates ROS levels in cells, and SLC38A1 (3-fold), whose expression is upregulated during OS [14]. CD also decreased the expression of SOD1 (2-fold), an internal antioxidant enzyme that regulates the level of oxidative stress caused by mitochondrial and cytosolic superoxide, and SOD2 (1.5-fold), an enzyme involved in cellular systems that repair oxidative damage [15]. These results are also consistent with those observed in the SOD experiments, where CD induced a decrease in SOD expression in the LPS-stimulated cells. Moreover, GSS, a gene crucial for several biological processes such as membrane transport, xenobiotic detoxification and cell defense against free radical oxidative damage [16], was significantly increased by CD treatment. The expression of GCLC mRNA, a crucial part of a long-term adaptive mechanism that enables cells to function under severe OS for extended periods [17], was also significantly increased by CD. Moreover, CAT was upregulated 2.9-fold, confirming the previous results with enzymatic assays. These results show that CD treatment increased the expression of antioxidant genes in response to LPS activation as part of the cells’ antioxidant defense. Pro-oxidant genes produce proteins commonly increased during inflammation and, through the inflammatory response, are also connected to OS [18]. Therefore, these results show CD’s ability to boost the cell’s antioxidant defense, indicating CD’s potent antioxidant properties.

The transcription of many cytoprotective genes is regulated by the transcription factor Nrf2, which is a key regulator of the redox balance and signaling and controls the expression of several antioxidant and detoxification genes by binding to antioxidant response elements (AREs). Low levels of Nrf2 signaling are an important contributor to many disease conditions associated with OS [19], regulating the cellular antioxidant response attached to the endogenous inhibitor Keap1. The Keap1 protein contains several cysteine residues with sulfhydryl groups that can react with ROS, causing the bonds between Nrf2 and Keap1 to break. Once the bonds are broken, Nrf2 becomes phosphorylated at Ser 40 and translocates to the nucleus of the cell, where it turns on the transcription of the gene coding for antioxidant enzymes. Nrf2-ARE binding controls the expression of more than 300 genes involved in cellular antioxidant and anti-inflammatory responses, such as CAT, SOD, GSH peroxidases, thioredoxin reductase, thioredoxin, sulfiredoxin, GSH S-transferase and glutamate-cysteine ligase [20,21]. In the current investigation, CD treatment upregulated 10-fold the Nrf2 mRNA expression compared to the control, and even in the presence of LPS, there was still an increase of 2-fold in the levels of Nrf2. The opposite effect was seen in the expression of Keap1, which was downregulated by CD, with or without the activation of the cells. The transcription and protein levels experiments confirmed that Nrf2-Keap1 signaling might be involved in the up-regulation of the antioxidant genes, including CAT and GSH, described in this study. Therefore, this investigation shows that the CD antioxidant mechanism may be mediated through the increasing levels of free Nrf2, which translocates to the nucleus and induces the transcription of many antioxidant and protective genes.

Moreover, a neurodegenerative disease study revealed that hippocampal astrocytes, one of the brain regions where neurodegeneration is first manifested in AD patients, had lower levels of Nrf2. At the same time, several studies show that NF-κB activation in astrocytes may worsen neuroinflammation and have neurotoxic effects [22,23,24,25]. Also, AD patients’ brains have enhanced NF-κB DNA-binding activity and expression of several NF-κB target genes. Many research investigations suggested that blocking NF-κB could be a key factor in stopping the progression of AD pathology. These findings highlight the significance of the study in terms of the interaction between the Nrf2 and NF-κB signaling pathways in AD progression [22,23,24,25]. In this study, the mRNA expression of numerous genes associated with NF-κB signaling was investigated using RT-PCR with specific primers. The objective was to examine if CD would downregulate the activation of NF-κB signaling. The findings showed that among the genes under investigation, CD downregulated mRNA expression of NF-κB1, NF-κB2 and IKBκB and increased mRNA expression of NFκBIA. Using specific antibodies against NF-κB1, NF-κB2, IKBκB and RELA, Western analysis was also performed to confirm PCR outcomes at the protein level. The results showed that there was an increase in the levels of NF-κB1, NF-κB2 and IKBκB when BV-2 microglial cells were treated with LPS, but that this induction was significantly reduced when CD was administered 1 h before LPS treatment, confirming the results from the RT-PCR assays and showing the neuroprotective effect of CD. The mRNA expression of RELA was not changed significantly either in the transcription or at the protein level.

CD downregulated mRNA expression of NF-κB1, the most frequently expressed transcription factor in macrophages and a crucial factor of chronic inflammation [23]. Studies show that mice lacking NF-κB1 develop persistent low-grade inflammation due to the loss of their ability to have inflammatory activity. Dysregulation of NF-κB signaling has been implicated in various inflammatory diseases, aging and life expectancy [24]. CD also downregulated the expression of NF-κB2, a gene that encodes an NF-κB subunit, part of the transcription factor complex and the primary activator of genes involved in immune response and inflammation [25]. Furthermore, CD downregulated IKBκB’s mRNA expression, which is a complex of enzymes and part of the NF-κB signaling pathway. This gene’s protein phosphorylates the inhibitor in the inhibitor/NF-κB complex, resulting in the inhibitor’s dissociation and NF-κB activation [26]. Moreover, CD increased the production of NFκBIA mRNA, which produces a protein that belongs to the family of NF-κB inhibitors, which blocks the nuclear localization signals of REL dimmers, trapping them in the cytoplasm and inhibiting the activity of dimeric NF-κB/REL complexes. Immune and proinflammatory reactions that stimulate cells cause them to become phosphorylated, encouraging ubiquitination and breakdown. This allows the dimeric RELA to go to the nucleus and trigger transcription. Transcriptional activators include the RELA-NF-κB1 and RELA-REL heterodimeric NF-κB complexes. It can affect the promoters’ accessibility to transcription factors in addition to serving as a direct transcriptional activator, which indirectly controls gene expression [27].

Because of its crucial function in forming inflammatory mediators, NF-κB has been considered a prime therapeutic target in models of neurodegenerative-disease-induced neurotoxicity. Nrf2 and NF-κB seem to interact in neurodegenerative illnesses, with Nrf2 acting as a neuroprotective factor while NF-κB worsens neuroinflammation [22]. The results of this study provide evidence that CD modulates genes and proteins associated with the activation of NF-κB. The data presented here are essential since the literature suggests that blocking NF-κB could be a key factor in stopping the progression of AD pathology and that increased levels of Nrf2 would reduce OS levels. Therefore, the CD modulatory effect on Nrf2 and NF-κB signaling observed in this study may be beneficial for preventing or halting the progression of neurodegenerative diseases such as AD.

In the present study, we used LPS-stimulated BV-2 microglial cells, which is a well-characterized, widely used model, especially for the investigation of neurodegenerative disorders involving immune responses, such as neuroinflammation and oxidative stress. Several articles have shown that BV-2 cells are suitable substitutes for primary microglia in many experimental settings, as well as in complex studies involving cell–cell interaction [28]. Studies comparing primary rat microglia to the BV-2 cell line observed that upon LPS stimulation, BV-2 cells secreted lesser but still substantial amounts of NO compared to primary microglia [29]. Henn et al. [28] investigated the BV-2 cells as an appropriate alternative to the primary cultures. They found that in response to LPS, 90% of genes induced in the BV-2 cells were also induced in primary microglia, indicating that this is a good research model. However, further studies will be needed using in vivo models to confirm and elucidate the molecular mechanisms that cardamonin uses in fighting oxidative stress and inflammation.

## 4. Materials and Methods

### 4.1. Materials

Genesee Scientific (San Diego, CA, USA) provided high glucose Dulbecco’s Modified Eagle’s Medium (DMEM), penicillin-streptomycin (10,000 U/mL), trypsin/EDTA (0.25%), phenol red and the heat-inactivated fetal bovine serum (HI-FBS). From Sigma-Aldrich Co., (St. Louis, MO, USA), we purchased dimethyl sulfoxide (DMSO), 99% pure cardamonin (C_16_H_14_O_4,_ Molecular Weight: 270.28) and *Escherichia coli* lipopolysaccharides (LPS). The cell-based assay kits for superoxide dismutase (SOD) (cat# 706002), catalase (CAT) (cat# 707002) and glutathione (GSH) (cat# 703002) were purchased from Cayman Chemical Co., (Ann Arbor, MI, USA). The materials for the oxidative stress PCR array (cat# 10034391) experiment and the RNase-Free DNase set, SsoAdvanced universal SYBR Green Supermix and primers were purchased from BioRAD Laboratories. (Hercules, CA, USA). Thermo Fisher Scientific Co., (Waltham, MA, USA) provided the BCA protein assay kit. The Western assay reagents and plates (cat# SM-W004) were all purchased from ProteinSimple (San Jose, CA, USA). Primary antibodies were purchased from Cell Signaling (Danvers, MA, USA). 

### 4.2. Cell Culture

The immortalized murine microglial BV-2 cell line was provided by Elisabeth Blasi’s group at the University of Perugia [30]. The BV-2 cells were cultured in 1% penicillin/streptomycin (100 U/mL penicillin and 0.1mg/mL streptomycin) high-glucose DMEM with 10% HI-FBS. (Genesee Scientific, San Diego, CA, USA). Cells were maintained at 37 °C in a humidified 5% CO_2_ environment, with the media changed every 2–3 days.

### 4.3. Cell Viability

In experimental cell culture conditions, the BV-2 cells were seeded (5 × 10^5^ cells/mL) in 96-well plates (100 µL/well) and exposed to CD at concentrations ranging from 6.25 to 200 µM for 24 h with and without activation with 1 µg/mL of LPS. To make the CD stock, we first dissolved it in DMSO and diluted it in an experimental medium to obtain the desired concentration. The control had a concentration of DMSO of no more than 0.02%. Resazurin dye (7-hydroxy-3H-Phenoxazin-3-one 10-oxide), a cell-permeable redox indicator used to evaluate the viable cell count, was utilized to determine cell viability. Resazurin was prepared as a working solution in HBSS (pH 7.4) at 0.5 mg/mL and then put into a sterilized, light-protected container after passing through a 0.2 µm filter. In the resazurin test, viable cells can convert the dark blue resazurin dye to the bright pink fluorescent resorufin product via redox reactions; the dye solution was added to the samples (20 µL of reagent to the final volume of 200 µL of the medium in the 96-well plate). After this, the samples were placed back in the incubator at 37 °C for 4 h. A microplate reader Infinite M200 (Tecan Trading AG, Männedorf, Switzerland). was used to quantitatively assess the conversion of resazurin to fluorescent resorufin, which is proportional to the amount of metabolically active, live cells present. The filter settings were 550 nm excitation/580 nm emission.

### 4.4. Measurement of Nitric Oxide (NO) Production

The effect of CD on NO production was studied in LPS-stimulated BV-2 cells. The released amount of nitrite, a soluble oxidation product of NO, was measured with a colorimetric assay using Griess reagent (1% sulfanilamide and 0.1% N-(1-naphthyl)-ethylenediamine dihydrochloride in 5% phosphoric acid (H_3_PO_4_)). The BV-2-cells (5 × 10^5^ cells/mL) in a 96-well plate were seeded overnight to attach the cells, then stimulated with LPS (1 µg/mL) in the presence or absence of CD (0.78–25 µM) for 24 h. Control cells were treated with DMSO. Fifty microliters of supernatant were mixed with an equal volume of the Griess reagent and were measured at 550 nm excitation and 580 nm emission wavelengths using a microplate reader Infinite M200 (Tecan Trading-AG, Mannedorf, Switzerland). Sodium nitrite was used as a standard to calculate the nitrate concentrations. Each set of standards, controls and samples had their mean absorbance calculated, from which the average optical density of the zero standards was deducted. In Excel (Microsoft 365), the standard concentration was used as the *x*-axis and absorbance was used as the *y*-axis to plot the standard curve. Through the predetermined locations, a best-fit straight line was created.

### 4.5. Glutathione, Catalase and Superoxide Dismutase Assays

#### 4.5.1. Cell Treatment

The BV-2 cells were seeded (5 × 10^5^ cells/mL) overnight in T-75 flasks in experimental cell culture media. The next day, the cells were treated with different treatments for 24 h, Control (Cells + DMSO), CD (6.25 µM), LPS 1 µg/mL and cotreatment (LPS 1 µg/mL and CD 6.25 µM). The CD stock was freshly prepared by initially dissolving it in DMSO and then diluting further with experimental media to the appropriate concentration for each treatment. The concentration of DMSO did not exceed 0.02%, which was used for the control. As previously reported, a cell scraper was used to harvest the cells that had been grown and treated in T-75 flasks. The cells were then collected by centrifuging them at 1000–2000× *g* for 10 min at 4 °C.

#### 4.5.2. Glutathione Assay

The pellet was homogenized using a sonicator in 1–2 mL of cold, 50 mM MES buffer at pH 6–7 and containing 1 mM EDTA. The mixture was then centrifuged at 10,000× *g* for 15 min at 4 °C. For deproteinization, the supernatants were kept on ice with freshly made 10% metaphosphoric acid (MPA) reagent in an equal volume. Triethanolamine (TEAM) was reacted with 53 µL of water to create a 4M solution, which was then added to the samples. Due to its 4-hour stability at 25 °C, the TEAM solution was newly made (50 µL of TEAM reagent was added for every ml of the supernatant) and the combination was vortexed right away. The TEAM reagent raised the pH of the sample. Fifty microliters of the sample or standard were added in duplicate (A/B) to each designated well on the plate. The plate was then covered, while the assay cocktail was freshly prepared for use within 10 min of preparation. The assay cocktail was prepared by mixing in a 20 mL vial of 11.25 mL MES Buffer, 0.45 mL reconstituted cofactor mixture, 2.1 mL reconstituted enzyme mixture, 2.3 mL water and 0.45 mL reconstituted DTNB. Immediately, 150 μL of the assay cocktail was added to each well for a total volume of 200 μL per well. The plate cover was replaced over the plate and the plate was incubated in the dark on an orbital shaker for 25 min at room temperature before measuring the absorbance in the wells at 410 nm using a microplate reader Infinite M200 (Tecan Trading AG-Mannedorf, Suica).

#### 4.5.3. Catalase Assay

The cell pellets were sonicated in 1–2 mL of cold buffer, pH 7.0, containing 1 mM EDTA and 50 mM potassium phosphate. The homogenized cells were then centrifuged at 10,000× *g* for 15 min at 4 °C. The supernatant was removed for assay and stored on ice. In all the wells, the final volume of the assay was 240 µL. To each designated well, 100 µL of diluted assay buffer, 3 µL of methanol and 20 µL of standard (tubes A–G) were added to the plate. In the positive control wells, a 100 µL of diluted assay buffer was added, 30 µL of methanol and 20 µL of diluted catalase (control) were added to two wells. In the sample wells, 100 µL of diluted assay buffer, 30 µL of methanol and 20 µL of the sample were added to two wells. The reactions were initiated by adding 20 µL of diluted hydrogen peroxide to all the wells being used. The precise time the reaction was initiated was noted and the diluted hydrogen peroxide was added as quickly as possible. The plate was then covered with the plate cover and incubated on a shaker for 20 min at room temperature. To terminate the reaction, 30 µL of potassium hydroxide was added to each well, and then 30 µL of catalase purpald was added to each well. The plate was then covered with the plate cover and incubated for 10 min on the shaker at room temperature. To each well, 10 µL of catalase potassium periodate was added and covered with the plate cover and incubated on a shaker for 5 min at room temperature before measuring the absorbance in the wells using a microplate reader Infinite M200 (Tecan Trading AG, Mannedorf, Switzerland) at 540 nm excitation and 580 nm emission filter settings.

#### 4.5.4. Superoxide Dismutase

The cell pellet was sonicated in cold 20 mM HEPES buffer (Sigma-Aldrich Co., St. Louis, MO, USA), pH 7.2, containing 1 mM EGTA, 210 mM mannitol and 70 mM sucrose. The homogenized cells were then centrifuged at 1500× *g* for five min at 4 °C. The supernatant was separated for the assay and stored on ice. The standards were prepared by diluting 20 µL of the SOD standard (Item No. 706005) with 1.98 mL of 1X sample buffer (diluted) to obtain the SOD stock solution. The dilute Radical Detector was prepared just before use by transferring 50 µL of the supplied solution to another vial, diluted with 19.95 mL of diluted assay buffer, and covered with tin foil. On the 96-well microplate, 200 µL of the diluted Radical Detector and 10 µL of SOD Standard or 10 µL of the sample were transferred to the designated wells on the plate. The reactions were initiated by adding 20 µL of diluted xanthine oxidase to all the wells. The plate was then carefully shaken for a few seconds to mix, covered and incubated on a shaker for 20 min at room temperature. The absorbance was measured at 440–460 nm using a microplate reader Infinite M200 (Tecan Trading AG, Mannedorf, Switzerland).

### 4.6. RT-PCR (Real-Time Polymerase Chain Reaction)

#### 4.6.1. RNA Isolation

The BV-2 cells were seeded (5 × 10^5^ cells/mL) overnight in T-75 flasks in experimental cell culture media. The next day, the cells were treated with different treatments for 24 h, control (cells + DMSO), CD (6.25 µM), LPS 1 µg/mL and cotreatment (LPS 1 µg/mL and CD 6.25 µM). The cells were harvested using a cell scraper and centrifuged at 1000 RPM for 3 min. If not used on the same day, the cell pellet was stored at −80 °C. In the first step of RNA isolation, we added 1 mL of Trizol to each sample and homogenized it for 30 s; in the second step, we added 20 µL of chloroform to each sample. The samples sat in the rack for 3–5 min at room temperature, vortexed samples for 15–30 s and centrifuged for 12 min at 11,000 RPM. The samples’ supernatant (clear layer) was added to a new tube in the hood area and combined with 500 µL of isopropanol alcohol for RNA precipitation. The samples were inverted in each tube gently 3 times and incubated at room temperature for 10 min, then centrifuged for 12 min at 11,000 rpm. The supernatant was removed and the RNA pellets were washed with 1mL of 75% ethanol; we inverted each sample 3 times and centrifuged for 5 min at room temperature. The RNA pellet was dissolved in 30 µL of RNase-free water, vortexed for 15 s and placed on ice for 30 min. Lastly, 2 µL of water and blank were added to the Nanodrop machine and the RNA concentration of samples was measured (A260/280 ratio should be around 2.0) to determine RNA concentration.

#### 4.6.2. cDNA Synthesis 

The cDNA strands were synthesized from the mRNA: a mixture of 4 µL of the 5X iScript advanced reaction mix containing dNTPs, Oligo(dT) and random primers, 1 µL of reverse transcriptase, 7.5 µL of water and 7.5 µL of the sample were combined in a 0.2 mL tube to make a total of 20 µL, and vortexed for 30 s. Samples were centrifuged at 1000 rpm for 1 min and placed inside the equipment. Reverse transcription was performed using the thermocycler program and the transcription steps were as follows: reverse transcription, 30 min at 42 °C and RT inactivation for 5 min at 85 °C. 

#### 4.6.3. Oxidative Stress RT-PCR Arrays

The manufacturer’s protocol for RT-PCR amplification was followed (BioRAD, Hercules, CA, USA). For the array assays, a 10 µL sample (10 ng cDNA/reaction) and 10 ul of the master mix were added to each well. Using the Bio-Rad CFX96 Real-Time System, the thermal cycling protocol included an initial hold step at 95 °C for 2 min and denaturation at 95 °C for 5 s, followed by 40 cycles of 60 °C for 30 s (annealing/extension), 60 °C for 5 s/step (melting curve) (Hercules, CA, USA). The primers employed in the PCR arrays were unique to each gene of interest and were designed to target a number of oxidative stress-related genes.

#### 4.6.4. RT-PCR with Individual Primers 

The real-time PCR amplification methodology recommended by the manufacturer was carried out as follows: for the individual primers, a mixture of 1 µL of the sample (200 ng cDNA/reaction), 10 µL of master mix, 1 µL of primer and 8 µL of water was added to each well. The thermal cycling protocol included an initial hold step at 95 °C for 2 min and denaturation at 95 °C for 15 s, followed by 40 cycles of 60 °C for 30 s (annealing/extension) and 60 °C for 5 s/step (melting curve) using the Bio-Rad CFX96 Real-Time System, (Hercules, CA, USA). The individual primers used were specific to each gene of interest. The Unique Assay ID for the primers is described as follows:

Nrf2 (UniqueAssayID: qMmuCID0021433); KEAP1 (UniqueAssayID: qMmuCID0008745); NFκBIA (UniqueAssayID: qMmuCED0045043); NF-κB1 (UniqueAssayID: qMmuCED0047222); NF-κB2 (UniqueAssayID: qMmuCED0040272); NIK (UniqueAssayID: qMmuCID0018436); IKBκB (UniqueAssayID: qMmuCID0005811); GAPDH (UniqueAssayID: qMmuCED0027497) RELA (UniqueAssayID: qMmuCID0017564).

### 4.7. Simple Western Assay 

This assay used automated Western Analysis to determine total proteins (ProteinSimple, San Jose, CA, USA). ProteinSimple provided all of the reagents and the analysis followed the user’s guide. Each loading protocol for an antibody or protein was optimized. An overall concentration of 0.2 mg/mL total protein, 1X sample buffer, 1X fluorescent molecular weight markers and 40 mM dithiothreitol was obtained by combining protein extracts with a master mix. Samples were heated for 5 min at 95 °C. Microplate wells were filled with the following reagents: samples, blocking solution, primary antibodies (dilution 1:5, 1:25, 1:125), horseradish peroxidase-conjugated secondary antibodies, chemiluminescent substrate and separation and stacking matrices. After plate loading, the capillary system performed completely automated electrophoresis and immunodetection. The secondary antibody linked to HRP and a chemiluminescent substrate were used to immunosorbent the proteins and these are the catalog numbers for the primary antibody that was utilized to identify the target proteins. The antibodies used are described below:

NRF2 (33649S); KEAP1 (4678S); NF-κB1 (13586S); IKBκB (8943S); NF-κB2 (52583S); RELA (8242S).

### 4.8. Data Analysis 

Graph Pad Prism was used for statistical analysis (version 6.07). The significance of the difference between the groups was determined using a one-way ANOVA, followed by a Dunnet multiple comparison test. * *p* ≤ 0.05, ** *p* ≤ 0.01, *** *p* ≤ 0.001, **** *p* ≤ 0.0001, ns = nonsignificant. All data were expressed as a mean, standard error from at least three independent trials. The CFX 3.1 Manager program was used to assess gene expression in the RT-PCR experiments. The ProteinSimple Compass program (version 6.2) was used to examine protein expression using capillary electrophoresis in Wes’ experiments.

## 5. Conclusions

This study shows CD’s great potential in reducing OS and oxidative damage by decreasing NO and increasing GSH levels and CAT activity. Furthermore, in the LPS-activated cells, CD reduced the mRNA expression of CCL5/Rantes, SLC38a1, NOS2, TXNIP, SOD1 and SOD2 while increasing the expression of GSS, GCLC and CAT, pointing out a reduction in inflammation and oxidative stress, as evidenced by PCR array profiling. In addition, CD increased the expression of Nrf2 and downregulated Keap1 expression, indicating that its antioxidant effects may be exerted via Nrf2/Keap1 signaling. Moreover, CD downregulated genes and proteins that are overproduced during inflammatory processes induced by the NF-kB signaling. These data corroborate with the literature that describes crosstalk between Nrf2 and NF-κB, where increased levels of Nrf2 could lead to neuroprotection against the harmful effects of NF-κB activation. In conclusion, these findings suggest that CD may decrease microglia-derived oxidative stress and inflammation and may prevent the onset or slow the progression of neurodegeneration caused by excessive oxidative stress and inflammatory processes in the CNS (Figure 13).

## Figures and Tables

**Figure 1 ijms-24-10872-f001:**
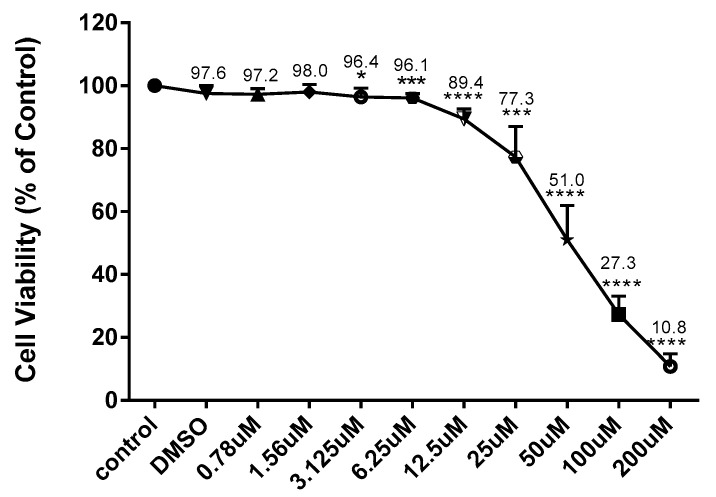
Effect of CD on the cell viability of BV-2 cells. Cell viability was evaluated using resazurin dye (7-hydroxy-3H-phenoxazin-3-one 10-oxide) 24 h after treatment. The data are presented as mean ± SEM and the significance of differences from the control treatment was determined using a one-way ANOVA and Dunnett’s multiple comparison tests. * *p* ≤ 0.05, *** *p* ≤ 0.001, **** *p* ≤ 0.0001.

**Figure 2 ijms-24-10872-f002:**
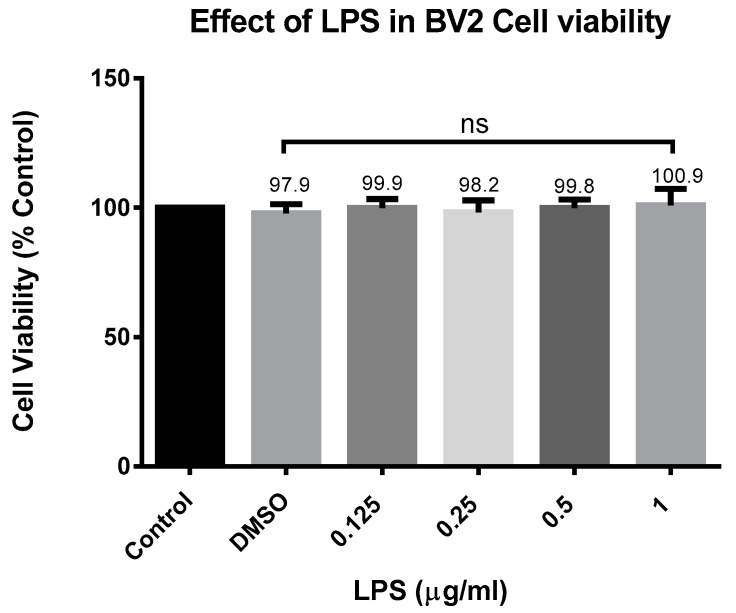
Effect of LPS on the cell viability of BV-2 cells. Cell viability was evaluated using resazurin dye (7-hydroxy-3H-phenoxazin-3-one 10-oxide) 24 h after treatment. The concentration of 1 µg/mL was used for further experiments. The data are presented as mean ± SEM and the significance of differences from DMSO treatment was determined using a one-way ANOVA and Dunnett’s multiple comparison tests. ns = nonsignificant.

**Figure 3 ijms-24-10872-f003:**
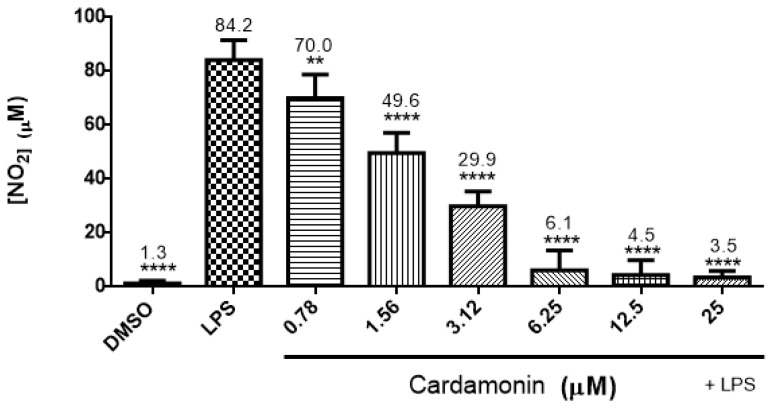
Effect of CD on nitric oxide production in LPS-activated BV-2 microglial cells. CD concentrations ranging from 0.78 µM to 25 µM were used. Nitric oxide production was measured using Griess reagent (1% sulfanilamide, 0.1% N-(1-naphthyl)-ethylenediamine hydrochloride in 5% phosphoric acid (H_3_PO_4_)) 24 h after treatment. The data are presented as mean ± SEM and the significance of differences from LPS treatment was determined using a one-way ANOVA and Dunnett’s multiple comparison tests. ** *p* ≤ 0.01, **** *p* ≤ 0.0001.

**Figure 4 ijms-24-10872-f004:**
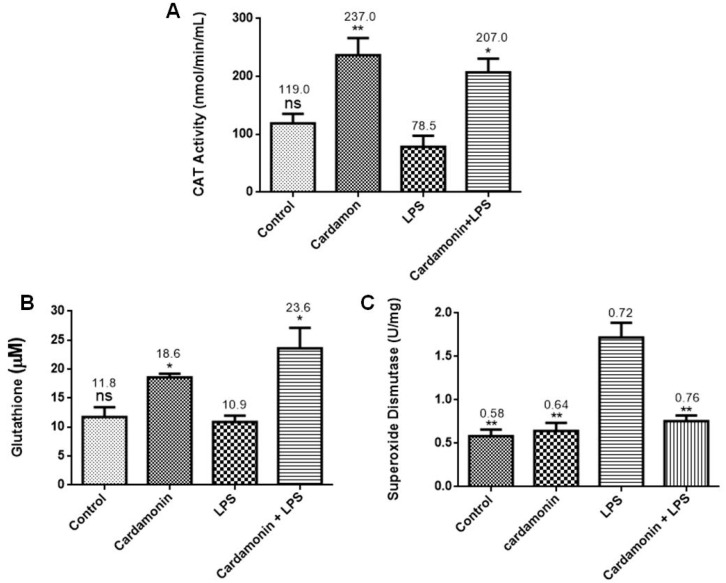
Effect of CD on the levels of CAT (**A**), GSH (**B**) and SOD (**C**) in BV-2 microglial cells. LPS 1 µg/mL stimulated the cells for 24 h. The data are presented as mean ± SEM and the significance of differences from LPS treatment was determined using a one-way ANOVA and Dunnett’s multiple comparison tests. * *p* ≤ 0.05, ** *p* ≤ 0.01, ns = nonsignificant.

**Figure 5 ijms-24-10872-f005:**
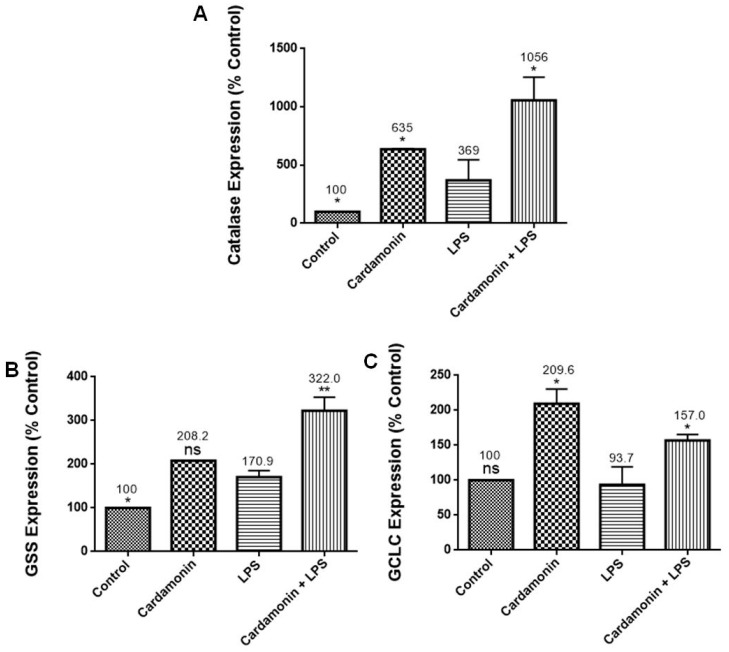
Effect of CD on the mRNA expression of CAT (**A**), GSS (**B**) and GCLC (**C**). The BV-2 microglial cells were stimulated with LPS 1 µg/mL for 24 h. The data are presented as mean ± SEM and the significance of differences from LPS treatment was determined using a one-way ANOVA and Dunnett’s multiple comparison tests. * *p* ≤ 0.05, ** *p* ≤ 0.01, ns = nonsignificant.

**Figure 6 ijms-24-10872-f006:**
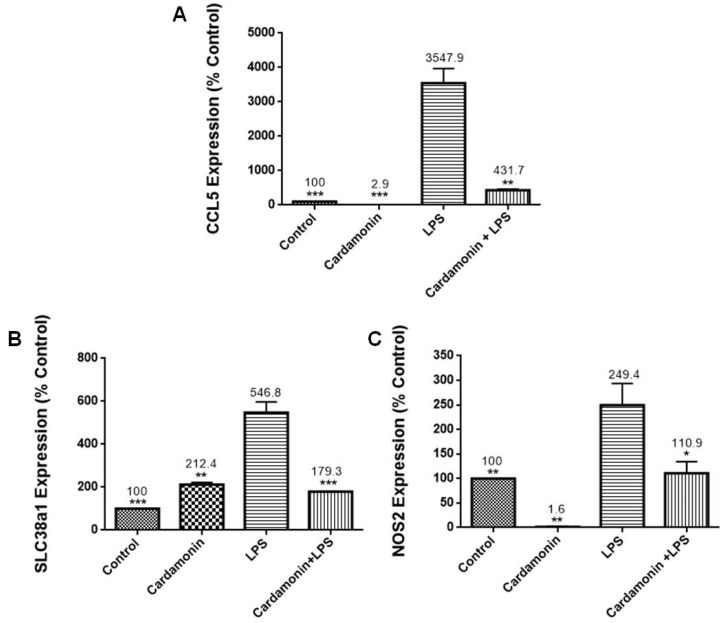
Effect of CD on the mRNA expression of CCL5/RANTES (**A**), SLC38a1 (**B**) and NOS2 (**C**). The BV-2 microglial cells were stimulated with LPS 1 µg/mL for 24 h. The data are presented as mean ± SEM and the significance of differences from LPS treatment was determined using a one-way ANOVA and Dunnett’s multiple comparison tests. * *p* ≤ 0.05, ** *p* ≤ 0.01, *** *p* ≤ 0.001.

**Figure 7 ijms-24-10872-f007:**
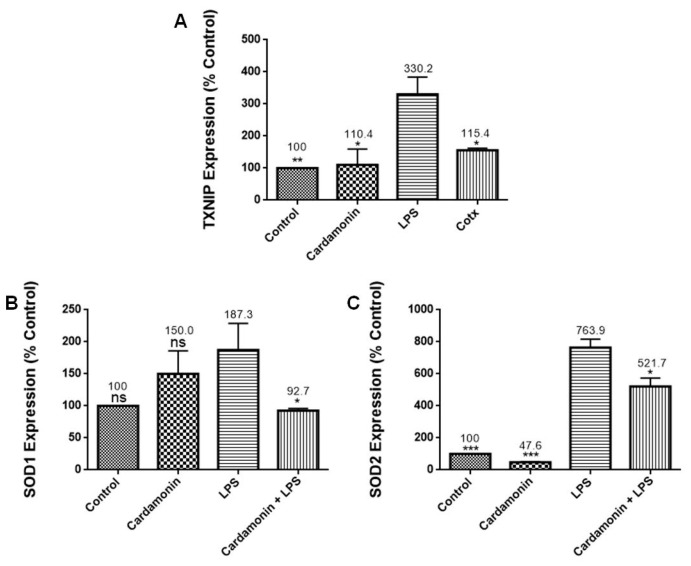
The effects of CD on the mRNA expression of TXNIP (**A**), SOD1 (**B**) and SOD2 (**C**). The BV-2 microglial cells were stimulated with LPS 1 µg/mL for 24 h. The data are presented as mean ± SEM and the significance of differences from LPS treatment was determined using a one-way ANOVA and Dunnett’s multiple comparison tests. * *p* ≤ 0.05, ** *p* ≤ 0.01, *** *p* ≤ 0.001, ns = nonsignificant.

**Figure 8 ijms-24-10872-f008:**
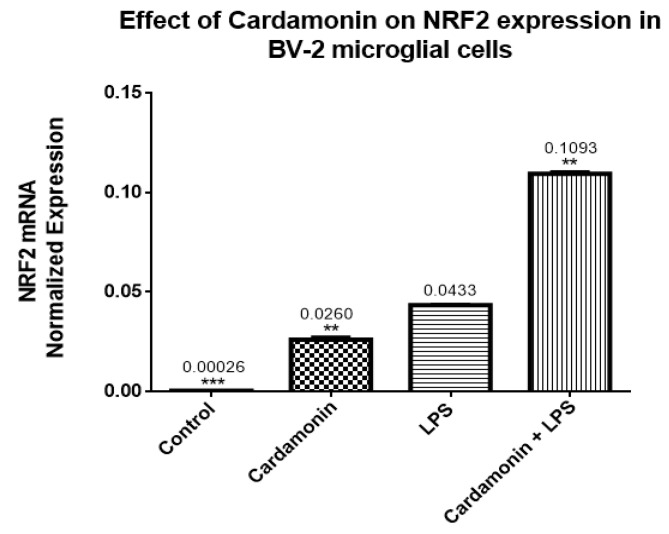
The effects of CD on the mRNA normalized expression of Nrf2. The BV-2 microglial cells were stimulated with LPS 1 µg/mL for 24 h. The data are presented as mean ± SEM and the statistical significance of the difference between LPS and different treatments was determined using a one-way ANOVA, followed by Dunnett’s multiple comparison tests ** *p* ≤ 0.01, *** *p* ≤ 0.001.

**Figure 9 ijms-24-10872-f009:**
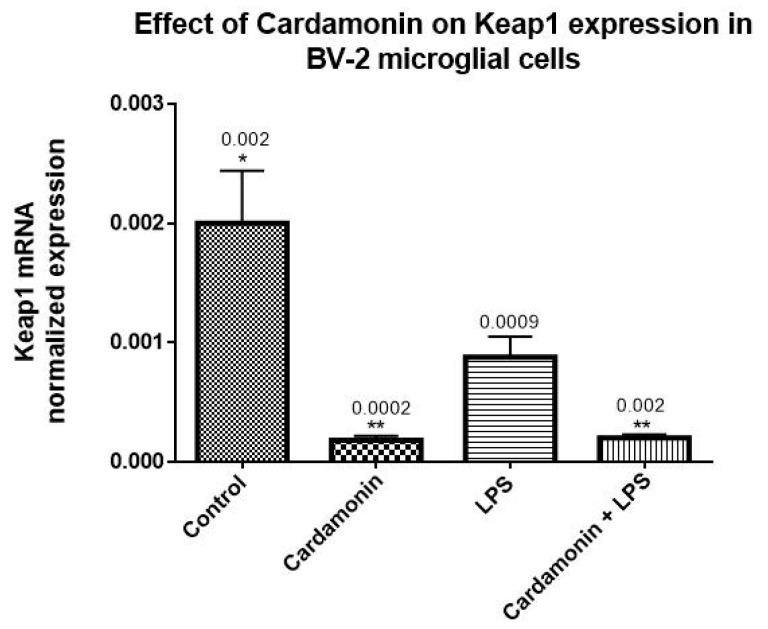
The effects of CD on the mRNA normalized expression of Keap1. The BV-2 microglial cells were stimulated with LPS 1 µg/mL for 24 h. The data are presented as mean ± SEM and the significance of the difference between LPS and different treatments was determined using a one-way ANOVA, followed by Dunnett’s multiple comparison tests. * *p* ≤ 0.05, ** *p* ≤ 0.01.

**Figure 10 ijms-24-10872-f010:**
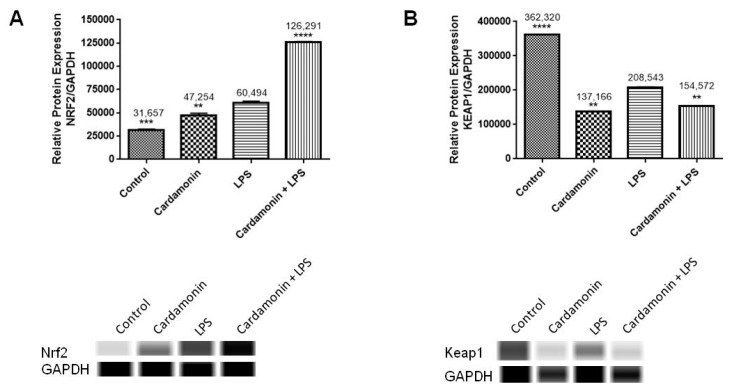
The effects of CD on the protein expression of Nrf2 (**A**) and Keap1 (**B**) in BV-2 microglial cells. The data are presented as mean ± SEM and the significance of differences from the LPS was determined using a one-way ANOVA and Dunnett’s multiple comparison tests. After 24 h of treatment, the bands show the control (DMSO), CD (6.25 µM), LPS and cotreated cells (CD + LPS). ** *p* ≤ 0.01, *** *p* ≤ 0.001, **** *p* ≤ 0.0001. GAPDH was used as the housekeeping gene.

**Figure 11 ijms-24-10872-f011:**
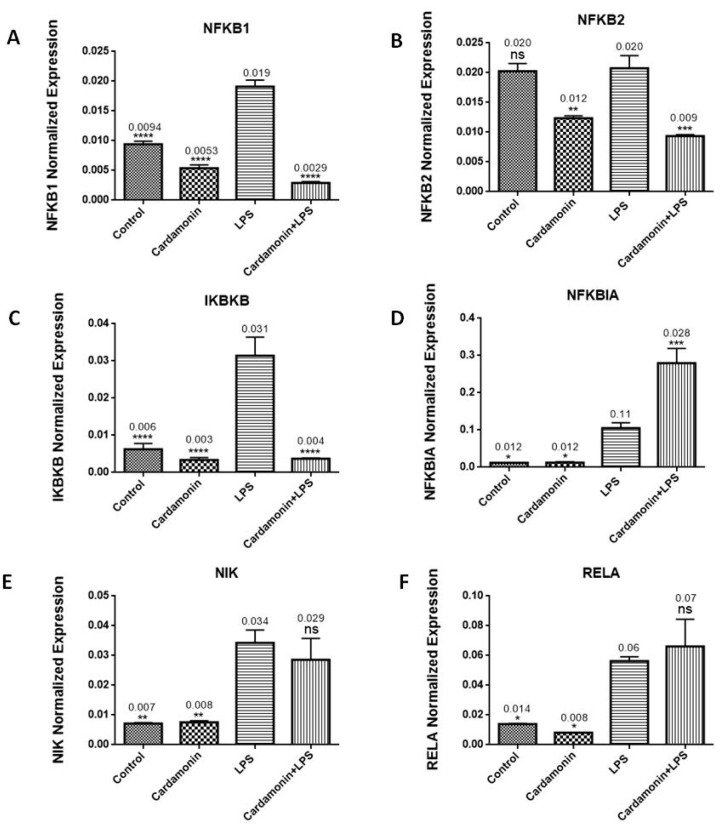
The effect of CD on the expression of genes involved in the NFkB signaling pathway: NF-κB1 (**A**), NF-κB2 (**B**), IKBκB (**C**), NFκBIA (**D**), NIK (**E**) and RELA (**F**). Graph bars show control, CD (6.25 µM), LPS and CD cotreatment (CD (1 h before) + LPS. Data represent normalized expression as the mean ± SEM, the statistical significance of the difference between LPS and different treatments was determined using a one-way ANOVA and Dunnett’s multiple comparison tests. * *p* ≤ 0.05, ** *p* ≤ 0.01 *** *p* ≤ 0.001, **** *p* ≤ 0.0001, ns = nonsignificant.

**Figure 12 ijms-24-10872-f012:**
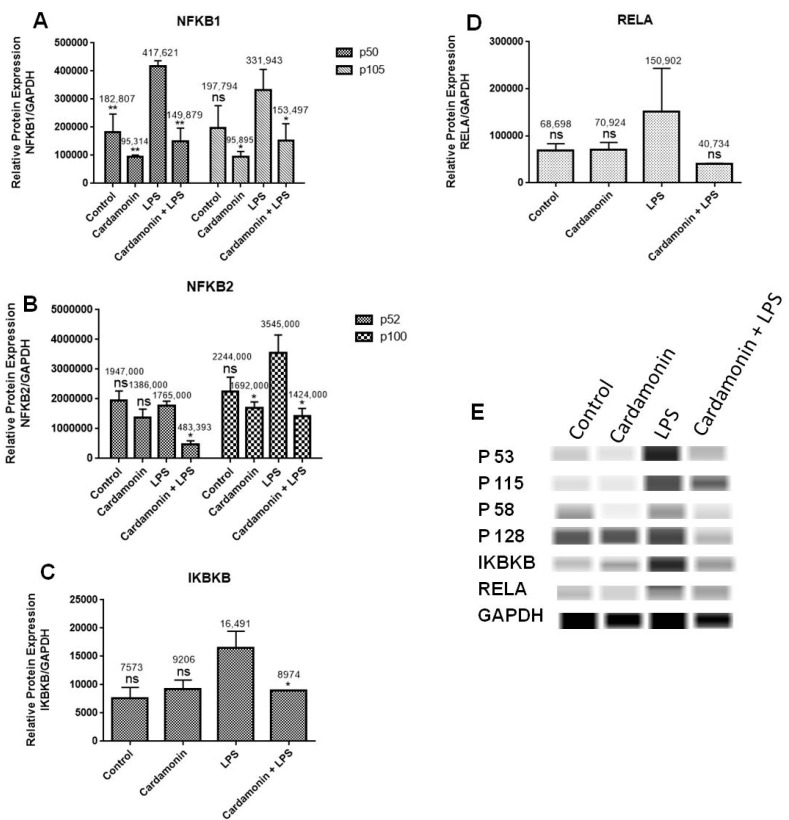
The effect of CD on the expression of proteins involved in the NF-kB signaling pathway: NF-κB1 (**A**), NF-κB2 (**B**), IKBκB (**C**), RELA (**D**) and bands (**E**) representing the protein expression after four treatments (24 h). Graph bars show control, CD (6.25 µM), LPS and CD cotreatment (CD 1 h before) + LPS. Data representing protein expression as the mean ± SEM and the statistical significance of the difference between LPS and different treatments were determined using a one-way ANOVA and Dunnett’s multiple comparison tests. * *p* ≤0.05, ** *p* ≤ 0.01, ns = nonsignificant.

**Figure 13 ijms-24-10872-f013:**
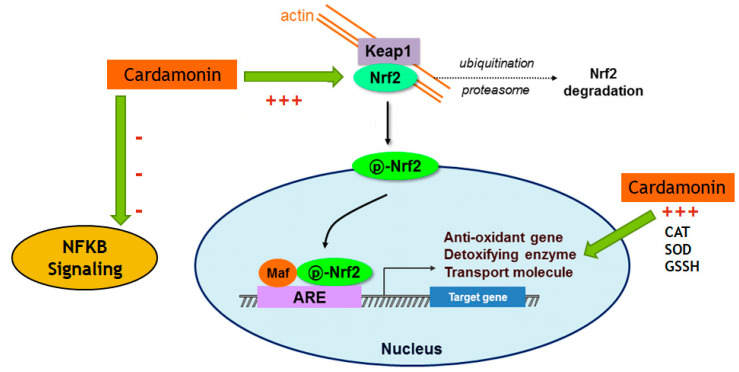
Proposed mechanism of CD molecular effects. Arrows indicate CD’s effects described in this study.

## Data Availability

Not applicable.

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
