# Peer review of "Antioxidant and Anti-Inflammatory Mechanisms of Cardamonin through Nrf2 Activation and NF-kB Suppression in LPS-Activated BV-2 Microglial Cells"

_ijms, 2023, doi:10.3390/ijms241310872_

Round 1
Reviewer 1 Report
Searching for natural products to combat neurodegenerative disease, especially Alzheimer's disease, is an important topic. However, several concerns exist, and wide revisions are required as follows:
1. Abstract: the background needs to be more concise. Also, clarify the tested range of concentrations of cardamonin.
2. Introduction:
- Lines 78-80: the paragraph begins with “Recent studies” but ends with one reference. Please revise.
- The writing style should be formal from the third-person perspective. Do not use we or our (E.g. line 85, “we examined CD's role” should be “The CD's role in ….was examined”).
3. The results section is considered a major drawback of the current study and needs extensive revisions and editing as follows:
- The authors should describe how much change was induced by LPS in the results compared with the control cells (i.e., % control). In addition, the degree of improvement in CD-treated cells compared to LPS-exposed cells.
- Throughout the results section, each paragraph should be directly followed by citing its figure. E.g. lines 12-138, CD significantly increased catalase activity (Fig. 4A) and glutathione (Fig. 4B) expression compared to the control group, but LPS did not show any statistically significant upregulation. However, the combination of CD and LPS induced a significant increase in catalase activity and glutathione expression compared to the control levels. The effects of CD on superoxide dismutase (Fig. 4C) expression in BV-2 microglial cells showed that CD had no significant effect on SOD production compared to the control group, but when cells were treated with LPS, there was a significant increase in SOD production. Treating the cells with CD and subsequent activation with LPS reduced the expression of SOD almost to control levels.
- The description of results in the text is contradicted by the figures in many parameters as follows:
• Lines 133-136: The effects of CD on superoxide dismutase expression in BV-2 microglial cells showed that CD had no significant effect on SOD production compared to the control group, but when cells were treated with LPS, there was a significant increase in SOD production. While in Figure 4C, the CD-treated group has the stars of significance, while the LPS group has no stars of significance.
• Line 149: We also observed a reduction in the expression of SLC38A1 (3-fold). However, in Fig. 6B, the CD-treated cells showed a significant (**) increase in SLC38A1 expression.
• The same for lines 235-236 and Fig 11. E and F.
Thus, it is highly needed to rewrite the results in a clear representative way after revising the text with the figures.
4. The discussion section in its present form is just a repetition of the introduction section and methods used with the mentioning of the results again. The authors should rewrite this section and give a detailed discussion of their findings with the possible interpretations and reduce the comparisons with the earlier studies. For instance, the authors have not clarified the possible causes for the opposing effect of CD on CAT and SOD. Moreover, in Lines 339-340, several studies should end with several references.
5. Material and methods:
- The authors should clarify the full detail of cardamonin (chemical formula, molecular weight, purity, producing company, city, and country).
- References of most method protocols are missed.
6. The sentence should not begin with an abbreviation like “CD" in lines 24 and 40. The verbs should be in the past, E.g. line 33, “show” should be “showed”.
7. Revise the reference formatting throughout the text E.g. lines 288 (13), 297 (14).
The sentence should not begin with abbreviations like “CD" in lines 24 and 40. The verbs should be in the past, E.g. line 33, “show” should be “showed”.
Author Response
Reviewer 1
- Abstract: the background needs to be more concise. Also, clarify the tested range of concentrations of cardamonin.
Response: The background was revised, and the different concentrations of cardamonin were clarified/added for each result described.
- - Lines 78-80: the paragraph begins with "Recent studies" but ends with one reference. Please revise.
Response: Reference was added.
- The writing style should be formal from the third-person perspective. Do not use we or our (E.g., line 85, "we examined CD's role" should be "The CD's role in ….was examined").
Response: Sentences were rewritten according to the reviewer's request.
- The authors should describe how much change was induced by LPS in the results compared with the control cells (i.e., % control). In addition, the degree of improvement in CD-treated cells compared to LPS-exposed cells.
Response: As requested, we updated the LPS cell viability graph (Figure 2) to show the differences between the control using just media only, the control using DMSO, and LPS concentrations from 0.125 to 1 μg/ml. All the treatments were compared to the control (media), and none were statistically significant. All the data points were included to clarify the effect of LPS in the concentrations tested.
The choice of 1ug/ml concentration for LPS was based on previous literature using the same concentration and the results of our cell viability, showing that this concentration is not toxic to the cells.
The cell viability using a range of concentrations of cardamonin had the objective of finding a specific concentration where at least 80% of the cells are alive. It would show an immune response after LPS stimulation. Based on this, the concentration of 6.25μM was chosen, where 96.1% of the cells were alive (Figure 1).
- Throughout the results section, each paragraph should be directly followed by citing its figure. E.g., in lines 12-138, CD significantly increased catalase activity (Fig. 4A) and glutathione (Fig. 4B) expression compared to the control group, but LPS did not show any statistically significant upregulation. However, the combination of CD and LPS induced a significant increase in catalase activity and glutathione expression compared to the control levels. The effects of CD on superoxide dismutase (Fig. 4C) expression in BV-2 microglial cells showed that CD had no significant effect on SOD production compared to the control group, but when cells were treated with LPS, there was a significant increase in SOD production. Treating the cells with CD and subsequent activation with LPS reduced the expression of SOD almost to control levels.
Response: Results Section was revised, and each paragraph is directly followed by citing its figure, as requested.
- The description of results in the text is contradicted by the figures in many parameters as follows: • Lines 133-136: The effects of CD on superoxide dismutase expression in BV-2 microglial cells showed that CD had no significant effect on SOD production compared to the control group, but when cells were treated with LPS, there was a significant increase in SOD production. While in Figure 4C, the CD-treated group has the stars of significance, while the LPS group has no stars of significance.
Response: The description of the results and the data on the graph are consistent. The treatment with cardamonin "only" and the "control group" showed no statistically significant difference. In Figure 4C, the CD group has the stars of significance because all the treatments are being compared to the LPS treatment and not to the control group, as explained in the subtitle of the figure. Therefore, the control group shows stars because it is being compared to LPS, and there is a statistically significant difference between the two treatments. To better clarify the description of the results, we added that "cardamonin only" had no significant effect on SOD production compared to the control. (Results Section: Effect of CD on Catalase, Glutathione, and Superoxide Dismutase Expression).
- Line 149: We also observed a reduction in the expression of SLC38A1 (3-fold). However, in Fig. 6B, the CD-treated cells showed a significant (**) increase in SLC38A1 expression. • The same for lines 235-236 and Fig 11. E and F. Thus, it is highly necessary to rewrite the results in a clear representative way after revising the text with the figures.
Response: We agree with the reviewer that the result description was incorrect. There was a decrease in the expression of SLC38A1, not with the treatment of "cardamonin only," but the decline happened when we added "cardamonin, and after 1h, the cells were stimulated with LPS", as observed in the graph. A clarification was added to the beginning of the description of the PCR array results: "For all the experiments, the cells were treated with CD for 1h and then stimulated by LPS. The results of these combined treatments showed that…" (Results section: "Effects of CD on Oxidative Stress PCR Arrays").
For the description of results for NF-kB signaling (lines 235-236), we also added the clarification: "The results showed that when cells were treated with CD and then stimulated by LPS after 1h, CD downregulated…" For figures 11E and F, we clarified in the description of the results that: "the combined treatment of CD + LPS did not cause any statistically significant effect" (Results section: "The Effect of on NF-kB Signaling-Associated Genes and Proteins").
- The discussion section in its present form is just a repetition of the introduction section and methods used with the mentioning of the results again. The authors should rewrite this Section, give a detailed discussion of their findings with possible interpretations, and reduce the comparisons with the earlier studies. For instance, the authors have not clarified the possible causes for the opposing effect of CD on CAT and SOD. Moreover, in Lines 339-340, several studies should end with several references.
Response: The discussion was revised as requested by the reviewer.
- Material and methods: - The authors should clarify the full details of cardamonin (chemical formula, molecular weight, purity, producing Company, city, and country).
Response: Information on purity and source Company was already described under the "Material and Methods Section," but we included chemical formula and molecular weight, as requested ("4. Materials and Methods").
- References of most method protocols are missed.
Response: Cell viability, Nitric Oxide, and PCR protocols were optimized in our lab and followed as described in the material and methods. Glutathione, catalase, and superoxide dismutase assays were performed according to the manufacturer's protocol. The catalog number for each assay was added to "4. Materials and Methods." PCR array assays followed Bio-Rad protocol and ProteinSimple Western Analysis according to ProteinSimple protocol. The catalog number for both assays was included in "4. Materials and Methods."
- The sentence should not begin with an abbreviation like "CD" in lines 24 and 40. The verbs should be in the past, e.g., in line 33, "show" should be "showed."
Response: All the sentences that started with abbreviations were modified, and the past tense of verbs was modified.
- Revise the reference formatting throughout the text, e.g., lines 288 (13), 297 (14).
Response: Formatting was corrected.
Comments on the Quality of English Language
The sentence should not begin with abbreviations like "CD" in lines 24 and 40. The verbs should be in the past, e.g., in line 33, "show" should be "showed."
Response: All the sentences that started with abbreviations were modified, and the past tense of verbs was modified.

Reviewer 2 Report
In this article, Barber et al. investigate the potential of cardamonin (CD) in reducing oxidative stress and inflammation in activated BV-2 microglial cells. The study demonstrates the potential of CD in reducing oxidative stress and inflammation in LPS-activated BV-2 microglial cells through Nrf2 activation and NF-kB suppression. Although the findings are interesting, the following concerns limit the enthusiasm in the current study:
- The paper focuses on the effects of cardamonin in BV-2 microglial cells, which are an immortalized murine cell line and do not completely recapitulate the effects one might expect in primary microglial cells. The authors need to show an effect on primary cells.
- The use of LPS to activate the microglia seems irrelevant physiologically unless you are targeting a specific disease.
- It would be important to test the effects of cardamonin in an in vivo setting to consider the complexity of the brain environment.
- While the paper identifies the Nrf2 activation and NF-kB suppression as potential molecular mechanisms underlying the effects of cardamonin, it does not show clear mechanisms of action of cardamonin.
- The authors also need to perform a dose-response experiment to identify a maximal dose with minimal toxicity.
- The graphs need to include individual data points on the bars and be color-blind friendly.
- The article has minor grammatical errors, including spelling, spacing, tenses, and punctuation. The authors are advised to proofread the article thoroughly.
Author Response
Reviewer 2
- The paper focuses on the effects of cardamonin in BV-2 microglial cells, which are immortalized murine cell lines and do not completely recapitulate the effects one might expect in primary microglial cells. The authors need to show an effect on primary cells.
Response: We appreciate the question and concern. Murine BV-2 microglial cells have been used widely across research, especially for investigating neurodegenerative disorders involving immune responses, such as neuroinflammation and oxidative stress. This is an alternative model to the low cell number and time-consuming techniques required to grow primary microglia cultures. BV-2 cells proliferate and are an excellent option to yield many cells quickly (1, 2).
BV-2 cells were developed by Blasi and colleagues (3) through retroviral transduction in 1990; since then, these cells have mainly been used. Studies comparing primary rat microglia to the BV-2 cell line observed that upon LPS stimulation, BV-2 cells secrete lesser but still substantial amounts of· NO compared to primary microglia (4). Henn et al. (5) investigated the BV-2 cells as an appropriate alternative to the primary cultures. They found that in response to LPS, 90% of genes induced in the BV-2 cells were also induced in primary microglia, indicating that this is a good research model.
The use of microglia cell lines speeds up research investigations. It decreases the need for continuous cell preparations and animal experimentation, provided that the cell line reproduces the in vivo situation of primary microglia. Based on this literature, our lab has been using this model for the last decade and publishing articles in peer-reviewed journals, indicating this model is appropriate to show the potential of natural compounds as neuroprotective agents against neurodegeneration (6, 7, 8).
References
1- Stansley, B., Post, J. & Hensley, K. A comparative review of cell culture systems for studying microglial biology in Alzheimer's disease. J Neuroinflammation 9, 115 (2012). https://doi.org/10.1186/1742-2094-9-115
2- Hou RC, Wu CC, Huang JR, Chen YS, Jeng KC. Oxidative toxicity in BV-2 microglia cells: sesamolin neuroprotection of H2O2 injury involving activation of p38 mitogen-activated protein kinase. Ann N Y Acad Sci. 2005 May;1042:279-85. doi 10.1196/annals.1338.050. PMID: 15965073.
3- Blasi E., Barluzzi R., Bocchini V., MAzzolla R., Bistoni F. Immortalization of murine microglial cells by a v-raf/v-myc carrying retrovirus. J Neuroimmunol. 1990;27(2-3):229-37.
4- Horvath R, McMenemy N, Alkaitis M, DeLeo J: Differential migration, LPS-induced cytokine, chemokine, and NO expression in immortalized BV-2 and HAPI cell lines and primary microglial cultures. J Neurochem 2008, 107:557–569.
5- Henn A, Lund S, Hedtjarn M, Schrattenholz A, Porzgen P, Leist M: The suitability of BV2 cells as an alternative model system for primary microglia cultures or animal experiments examining brain inflammation. ALTEX 2009, 26:83–94.
6- Hilliard A, Mendonca P, Soliman KFA. Involvement of NFƙB and MAPK signaling pathways in the preventive effects of Ganoderma lucidum on the inflammation of BV-2 microglial cells induced by LPS. J Neuroimmunol. 2020 Aug 15;345:577269. doi: 10.1016/j.jneuroim.2020.577269. Epub 2020 May 26. PMID: 32480240; PMCID: PMC7382303.
7- Mendonca P, Taka E, Bauer D, Reams RR, Soliman KFA. The attenuating effects of 1,2,3,4,6 penta-O-galloyl-β-d-glucose on pro-inflammatory responses of LPS/IFNγ-activated BV-2 microglial cells through NFÆ™B and MAPK signaling pathways. J Neuroimmunol. 2018 Nov 15;324:43-53. doi: 10.1016/j.jneuroim.2018.09.004. Epub 2018 Sep 11. PMID: 30236786; PMCID: PMC6245951.
8- Cobourne-Duval MK, Taka E, Mendonca P, Soliman KFA. Thymoquinone increases the expression of neuroprotective proteins while decreasing the expression of pro-inflammatory cytokines and the gene expression NFκB pathway signaling targets in LPS/IFNγ -activated BV-2 microglia cells. J Neuroimmunol. 2018 Jul 15;320:87-97. doi: 10.1016/j.jneuroim.2018.04.018. Epub 2018 May 4. PMID: 29759145; PMCID: PMC5967628.
- The use of LPS to activate the microglia seems irrelevant physiologically unless you are targeting a specific disease.
Response: BV-2 microglial cells activated by LPS were the model we chose to induce the immune response we needed to have inflammation and oxidative stress. The model allowed us to measure the increased levels of gene and protein expression in these two processes, allowing us to determine the anti-inflammatory and anti-oxidative effects of cardamonin.
- It would be important to test the effects of cardamonin in an in vivo setting to consider the complexity of the brain environment.
Response: We agree with the reviewer that it is important to investigate the effects of cardamonin in an in vivo setting, and this is part of our future project. The aim of the present study was to show the potential of cardamonin to combat inflammation and oxidative stress observed in many neurodegenerative diseases.
- While the paper identifies the Nrf2 activation and NF-kB suppression as potential molecular mechanisms underlying the effects of cardamonin, it does not show clear mechanisms of action of cardamonin.
Response: The present work showed that cardamonin could increase the expression of catalase and glutathione, potent antioxidants in the body. Their transcription is induced by the activation of Nrf2, a transcription factor released from the Keap1/Nrf2 complex in the presence of oxidative stress. By testing the effect of cardamonin on Nrf2 mRNA and protein levels, we observed an increased expression. These data indicate that the mechanism of action of cardamonin would be by activating Nrf2, which translocates to the nucleus of the cell and, consequently, induces the expression of more than 300 antioxidants genes, including catalase and glutathione. How cardamonin enters the cells and causes, Nrf2 dissociation from the Keap1 complex is not clear yet. Therefore, with our assays, we can suggest a direct effect of cardamonin on Nrf2 expression and an indirect effect on glutathione and catalase levels induced by Nrf2 activation.
Moreover, cardamonin showed a down-regulatory effect on nitric oxide production and reduced the mRNA expression of NOS2. Considering that the activation of NF-kB signaling induces NO and NOS2, we investigated the effect of cardamonin on the expression of genes and proteins involved in the NF-kB signaling activation. The data show a reduction in the expression of genes and proteins of NF-kB signaling, which would inhibit the translocation of NF-kB to the nucleus of the cell and, consequently, reduce the levels of NO and NOS2. So, the present work gives evidence that the mechanism of action of cardamonin is through the inhibition of NRF2 activation and via NF-kB inhibition, which would increase levels of antioxidants and decrease inflammatory mediators, as observed in our results.
- The authors also need to perform a dose-response experiment to identify a maximal dose with minimal toxicity.
Response: Figure number 1 shows the dose-response effect of cardamonin on BV-2 cell viability in the range concentration of 0.78 – 200μM. This assay aimed to find a concentration of cardamonin that would stimulate the cells without killing more than 80% of them. That is the reason why we selected the concentration of 6.25μM. In this concentration, 96.1% of the cells are alive, and there are no statistical significance differences between 6.25μM and the lower concentrations tested in this study.
- The graphs must include individual data points on the bars and be color-blind friendly.
Response: All the graphs were updated to have individual data points and data bars that are color-blind friendly.
- The article has minor grammatical errors, including spelling, spacing, tenses, and punctuation. The authors are advised to proofread the article thoroughly.
Response: We appreciate the observation, and the article was thoroughly revised.

Round 2
Reviewer 1 Report
No further comments are to be addressed
none
Author Response
No changes needed

Reviewer 2 Report
The authors have addressed a few comments; however, the issue of primary cells and in vivo studies still persists. I would recommend the authors to add a detailed caveats section in the discussion addressing these concerns.
Author Response
Dear Editor:
We are pleased to resubmit the revised Manuscript ID: nutrients- 2459263. Title: “Antioxidant and Anti-inflammatory Mechanisms of Cardamonin through Nrf2 Activation and NF-kB Suppression in LPS-Activated BV-2 Microglial Cells”. We addressed the reviewer’s number 2 concerns as outlined below.
Reviewer 2
Comment: The authors have addressed a few comments; however, the issue of primary cells and in vivo studies still persists. I would recommend the authors add a detailed caveats section in the discussion addressing these concerns.
Response: As requested by reviewer number 2, we included information on using BV-2 microglial cells, primary microglia, and the need for future studies, including in vivo experiments.
The addition can be found in the last paragraph of the discussion and is described as follows:
“In the present study, we used LPS-stimulated BV-2 microglial cells, which is a well-characterized, widely used model, especially for investigating neurodegenerative disorders involving immune responses, such as neuroinflammation and oxidative stress. Several articles have shown that BV-2 cells are suitable substitutes for primary microglia in many experimental settings and in complex studies involving cell-cell interaction [29]. Studies comparing primary rat microglia to the BV-2 cell line observed that upon LPS stimulation, BV-2 cells secreted lesser but still substantial amounts of NO compared to primary microglia [30]. Henn et al. [29] investigated the BV-2 cells as an appropriate alternative to the primary cultures. They found that in response to LPS, 90% of genes induced in the BV-2 cells were also induced in primary microglia, indicating that this is a good research model. However, further studies will be needed using in vivo models to confirm and elucidate the molecular mechanisms cardamonin uses in fighting oxidative stress and inflammation.”
